# MLPerf Tiny Benchmark

Colby Banbury[*] Vijay Janapa Reddi[*] Peter Torelli[†] Jeremy Holleman[‡ ‖] Nat Jeffries[§]

Csaba Kiraly[¶] Pietro Montino[⋆] David Kanter[**] Sebastian Ahmed[††] Danilo Pau[‡‡]

Urmish Thakker[I] Antonio Torrini[II] Pete Warden[§] Jay Cordaro[‡] Giuseppe Di Guglielmo[III]

Javier Duarte[IV] Stephen Gibellini[‡] Videet Parekh[V] Honson Tran[V] Nhan Tran[VI]

Niu Wenxu[VII] Xu Xuesong[VII]

## Abstract

Advancements in ultra-low-power *tiny* machine learning (TinyML) systems promise to unlock an entirely new class of smart applications. However, continued progress is limited by the lack of a widely accepted and easily reproducible benchmark for these systems. To meet this need, we present MLPerf Tiny, the first industry-standard benchmark suite for ultra-low-power tiny machine learning systems. The benchmark suite is the collaborative effort of more than 50 organizations from industry and academia and reflects the needs of the community. MLPerf Tiny measures the accuracy, latency, and energy of machine learning inference to properly evaluate the tradeoffs between systems. Additionally, MLPerf Tiny implements a modular design that enables benchmark submitters to show the benefits of their product, regardless of where it falls on the ML deployment stack, in a fair and reproducible manner. The suite features four benchmarks: keyword spotting, visual wake words, image classification, and anomaly detection.

## 1 Introduction

Machine learning (ML) inference on the edge is an increasingly attractive prospect due to its potential for increasing energy efficiency [4], privacy, responsiveness, and autonomy of edge devices. Thus far, the field edge ML has predominantly focused on mobile inference, but in recent years, there have been major strides towards expanding the scope of edge ML to ultra-low-power devices. The field, known as "TinyML" [1], achieves ML inference under a milliWatt, and thereby breaks the traditional power barrier preventing widely distributed machine intelligence. By performing inference on-device, and near-sensor, TinyML enables greater responsiveness and privacy while avoiding the energy cost associated with wireless communication, which at this scale is far higher than that of compute [5]. Furthermore, the efficiency of TinyML enables a class of smart, battery-powered, always-on applications that can revolutionize the real-time collection and processing of data. Deploying advanced ML applications at this scale requires the co-optimization of each layer of the ML deployment stack to achieve the maximum efficiency. Due to this complex optimization, the

---

[*]Harvard University, [†]EEMBC, [‡]Syntiant [‖]UNC Charlotte [§]Google [¶]Digital Catapult [⋆]VoiceMed [**]MLCommons [††]Qualcomm [‡‡]STMicroelectronics [I]SambaNova Systems [II]Silicon Labs [III]Columbia [IV]UCSD [V]Latent AI [VI]Fermilab [VII]Peng Cheng Labs

35th Conference on Neural Information Processing Systems (NeurIPS 2021) Track on Datasets and Benchmarks.

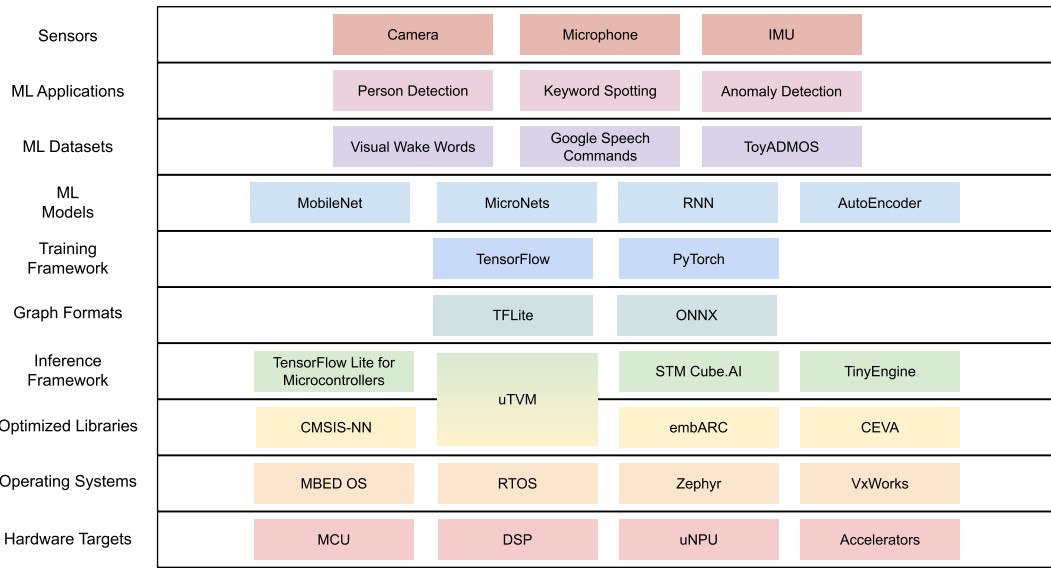

Figure 1: Summary of the Tiny Machine Learning Stack. There is rich diversity at every level of the maching learning computing stack, which makes standardization for benchmarking challenging.

direct comparison of solutions is challenging and the impact of individual optimizations is difficult to measure. In order to enable the continued innovation, a fair and reliable method of comparison is needed.

In this paper, we present MLPerf Tiny, an open-source benchmark suite for TinyML systems. The MLPerf Tiny inference benchmark suite provides a set of four standard benchmarks, selected by more than 50 organizations in academia and industry. In order to capture the tradeoffs inherent to TinyML, the benchmark suite measures latency, energy, and accuracy. MLPerf Tiny is designed with flexibility and modularity in mind to support hardware and software users alike and provides complete reference implementations to act as open-source community baselines.

## 2 Challenges

TinyML systems present a number of unique challenges to the design of a performance benchmark that can be used to measure and quantify performance differences between various systems systematically. We discuss the four primary obstacles and postulate how they might be overcome.

**Low Power** Power consumption is one of the defining features of TinyML systems. Therefore, a useful benchmark should profile the energy efficiency of each device. However, there are many challenges in fairly measuring energy consumption. TinyML devices can consume drastically different amounts of power,which makes maintaining accuracy across the range of devices difficult. Also, determining what falls under the scope of the power measurement is difficult to determine when data paths and pre-processing steps can vary significantly between devices. Other factors like chip peripherals and underlying firmware can impact the measurements.

**Limited Memory** While traditional ML systems like smartphones cope with resource constraints in the order of a few GBs, tinyML systems are typically coping with resources that are two orders of magnitude smaller. Traditional ML benchmarks use inference models that have drastically higher peak memory requirements (in the order of gigabytes) than TinyML devices can provide. This also complicates the deployment of a benchmarking suite as any overhead can make the benchmark too big to fit. Individual benchmarks must also cover a wide range of devices; therefore, multiple levels of quantization and precision should be represented in the benchmarking suite. Finally, a variety of benchmarks should be chosen such that the diversity of the field is supported.

**Hardware Heterogeneity** Despite its nascency, TinyML systems are already diverse in their performance, power, and capabilities. Devices range from general-purpose MCUs to novel architectures, like in event-based neural processors [2] or memory compute citekim20191. This heterogeneity poses

a number of challenges as the system under test (SUT) will not necessarily include otherwise standard features, like a system clock or debug interface. Furthermore, creating a standard interface while minimizing porting effort is a key challenge. Today's state-of-the-art benchmarks are not designed to handle these challenges readily. They need reengineering to be flexible enough to handle the hardware heterogeneity that is commonplace in the TinyML ecosystem.

**Software Heterogeneity** TinyML systems are often tightly coupled with their inference stack and deployment tools. To achieve the highest efficiency, users often develop their own tool chains to optimally deploy and execute a model on their hardware systems. This becomes increasingly critical on systems with multiple compute units, like accelerators and DSPs. This poses a challenge when designing a benchmark for these systems because any restriction, posed by the benchmark, on the inference stack, would negatively impact performance on these systems and result in unrepresentative results. Therefore we must balance optimality with portability, and comparability with representativeness. A TinyML benchmark should support many options for model deployment and not impose any restrictions that may unfairly impact users with a different deployment stack.

**Cross-product** Figure 1 illustrates the diversity of options at every level in the TinyML stack. Each option and each layer has its own impact on performance and TinyML software users can provide an improvement to the overall system at any layer. A TinyML benchmark should enable these users to demonstrate the performance benefits of their solution in a controlled setting.

# 3 Related Work

There are a few ML related hardware benchmarks, however, none that accurately represent the performance of TinyML workloads on tiny hardware.

**EEMBC(r)'s CoreMark(r)** benchmark [9] has become the standard benchmark for MCU-class devices due to its ease of implementation and use of real algorithms. Yet, CoreMark does not profile full programs, nor does it accurately represent machine learning inference workloads.

**EEMBC's MLMark(r)** benchmark [19] addresses these issues by using actual ML inference workloads. However, the supported models are far too large for MCU-class devices and are not representative of TinyML workloads. They require far too much memory (GBs) and have significant runtimes. Additionally, while CoreMark supports power measurements with the EEMBC ULPMark(tm)-CM benchmark, MLMark does not, which is critical for a TinyML benchmark.

**MLPerf**, a community-driven benchmarking effort, has recently introduced a benchmarking suite for ML inference [18] and has plans to add power measurements. However, much like MLMark, the current MLPerf inference benchmark precludes MCUs and other resource-constrained platforms due to a lack of small benchmarks and compatible implementations. As Table 1 summarizes, there is a clear and distinct need for a TinyML benchmark that caters to the unique needs of ML workloads, makes power a first-class citizen and prescribes a methodology that suits TinyML.

# 4 Benchmarks

All machine learning benchmarks fall somewhere on the continuum between low level and application level evaluation. Low level benchmarks target kernels that are core to many ML workloads, like matrix multiply, but they gloss critical elements like memory bandwidth or model level optimizations. On the other hand, application level benchmarks can obscure the target of the benchmark behind other stages of the application pipeline. MLPerf Tiny specifically targets model inference and does not include pre- or post-processing in the measurement window. Table 1 shows the v0.5 benchmarks.

In this section, we describe the benchmarks that form the MLPerf Tiny benchmark suite. Each benchmark targets a specific usecase and specifies a dataset, model, and quality target. Additionally, each benchmark has a reference implementation that includes training scripts, pre-trained models, and C code implementations. The reference implementations run the reference models in the TFLite format using TFLite for Microcontrollers (TFLM) [7] on the NUCLEO-L4R5ZI board. A known-good snapshot of the TFLM runtime is used to ensure stability, and the reference implementation is built using a bare-metal MBED project with the GCC-ARM toolchain. The reference implementations are open source and available on GitHub at `https://github.com/mlcommons/tiny`

| Use Case | Dataset (Input Size) | Model (TFLite Model Size) | Quality Target (Metric) |
|---|---|---|---|
| Keyword Spotting | Speech Commands (49x10) | DS-CNN (52.5 KB) | 90% (Top-1) |
| Visual Wake Words | VWW Dataset (96x96) | MobileNetV1 (325 KB) | 80% (Top-1) |
| Image Classification | CIFAR10 (32x32) | ResNet (96 KB) | 85% (Top-1) |
| Anomaly Detection | ToyADMOS (5*128) | FC-AutoEncoder (270 KB) | .85 (AUC) |

Table 1: MLPerf Tiny v0.5 Inference Benchmarks.

## 4.1 Visual Wake Words

**Rational** Tiny image processing models are becoming increasingly widespread, primarily for simple image classification tasks. The Visual Wakewords challenge [6] tasked submitters with detecting whether at least one person is in an image. This task is directly relevant to smart doorbell and occupancy applications, and the reference network provided as part of the challenge fits on most 32-bit embedded microcontrollers.

**Dataset** The Visual Wakewords Challenge uses the MSCOCO 2014 dataset [15] as the training, validation and test datasets for all person detection models. The dataset is preprocessed to train on image which contain at least one person occupying more than 2.5% of the source image. Images are also resized to 96x96 for model training.

**Model** The Visual Wakewords model is a MobilenetV1 [11] which takes 96x96 input images with an alpha of 0.25 and two output classes (person and no person). The TFLM model is 325KB in size.

**Quality Target** Based on validation and training accuracy, the model reaches about 86% accuracy across the preprocessed MSCOCO 2014 test dataset. In order to accommodate changes in accuracy due to quantization and rounding differences between platforms, submissions to the closed category should reach at least 80% accuracy across the same dataset.

## 4.2 Image Classification

**Rational** Novel machine vision techniques offer a cross-industry potential for breakthroughs and advances in autonomous and low power embedded solutions. Compact vision systems performing image classification at low cost, high efficiency, low latency and high performances are pervading manufacturing, IoT devices, and autonomous agents and vehicles. New hardware platforms, algorithms and development tools form a wide variety of deep embedded vision systems requiring a point of reference for scientific and industrial evaluation.

**Dataset** CIFAR-10 [14] is a labeled subset of the 80 Million Tiny Images dataset [20]. The low resolution of the images make CIFAR-10 the most suitable source of data for training tiny image classification models. It consists of 60000 32x32x3 RGB images, with 6000 images per class. The 10 different classes represent airplanes, cars, birds, cats, deers, dogs, frogs, horses, ships and trucks. The dataset is divided into five training batches and one testing batch, each with 10000 images. A significant amount of prior work in TinyML has used CIFAR-10 as a target dataset [8], by continuing this trend, we create a point of reference in the benchmark suite that can be used to relate future results to historical data points.

**Model** The Image Classification model is a customized ResNetv1 [10] which takes 32x32x3 input images and outputs a probability vector of size 10. The custom model is made of fewer residual stacks than the official ResNet: three compared to four. Moreover, the first convolutional layer is not followed by the pooling layer due to the low resolution of the input data. The number of convolution filters and the convolution strides dimension are lower as well compared to the official ResNet. The TFLite model for IC is 96KB in size and fits on most 32-bit embedded microcontrollers.

**Quality Target** A set of 200 images from the CIFAR-10 test set are selected to evaluate the performances of the IC reference implementation. The performance evaluation test is performed with the benchmark framework software, i.e. the runner (see Appendix). The model reaches 86.5% accuracy

across the 200 testing raw images. To accommodate minor differences in quantization and various other optimizations, we set the quality target to 85% top-1 accuracy.

## 4.3 Keyword Spotting

**Rational** Recognition of specific words and brief phrases, known as keyword spotting, is one of the primary use cases of ultra-low-power machine learning. Voice is an important mode of human-machine interaction. Wakeword detection, a specific case of keyword spotting wherein a detector continuously monitors for a specific word or phrase (e.g. "Hey Siri"™, "Hey Google"™, "Alexa"™) in order to enable a larger processor, requires continuous operation and therefore low power consumption. For example, a 100 mA current drain would deplete a typical phone battery in one day without any other activity. Command phrases such as "volume up" or "turn left" are the provide a simple and natural interface to embedded devices. Both cases require low latency and low power consumption and must typically run on small, low-cost devices.

**Dataset** We used the Speech Commands v2 dataset[21], a collection of 105,829 utterances collected from 2,618 speakers with a variety of accents. It is freely available for download under the Creative Commons BY license. The dataset contains 30 words and a collection of background noises and is divided into training, validation, and test subsets such that any individual speaker only appears in one subset. Following the typical usage of this dataset, we used 10 words and combined the background noise with the remaining 20 words to approximate an open set labeled "unknown," which, along with "silence", results in 12 output classes. This approach exercises a combination command-phrase systems' need for multiple words with wakeword systems' need for an open set of background noise.

**Model** For the closed division model, we used the small depthwise-separable CNN described in [22]. We chose this model because with 38.6K parameters, it fits within the available memory of most microcontrollers and similarly-scaled devices while achieving accuracy of 92.2% in our experiments. The model utilizes standard layers that can be expected of most neural network hardware. For this version of the benchmark, we opted to exclude feature extraction from the measurement and provide three choices of pre-computed features for the open division. Feature extraction is typically a small fraction of the overall compute cost, so the impact on the measurements is minor. While the pre-selected feature choices somewhat limit innovation in the overall KWS system, the features chosen represent the features most commonly used in KWS systems.

**Quality Target** To evaluate accuracy on the device under test, we randomly selected 1000 utterances from the speech commands. The quantized reference model demonstrated 91.6% accuracy on the full test set and 91.7% on the 1000-utterance subset. To allow for slight variations in quantization strategies, we set an accuracy requirement of 90%.

## 4.4 Anomaly Detection

**Rational** Anomaly detection is broadly the task of separating normal samples from anomalous samples and has applications in a large number of fields. It's unsupervised variant, which is used in our benchmark, is of particular importance to industrial use cases such as early detection of machine anomalies, where failure types are many and failure events can be rare, and thus data is often only available on normal operation for training. Anomalies can be observed in various data sources (or combinations thereof), including power, temperature, vibration, with the most intuitive being the detection of anomalies based on sound captured by microphones. The most important distinctive features of this benchmark are the use of unsupervised learning and an autoencoder (AE).

**Dataset** For this benchmark we have selected to use the dataset from the DCASE2020 [13] competition, which is itself a combination of two publicly available sources: the ToyADMOS [12], and the MIMII [17] datasets. The DCASE dataset contains data for six machine types (slide rail, fan, pump, valve, toy-car, toy-conveyor), and competition rules allowed for the training of separate models for each. Benchmarking would not benefit from this type of complexity, thus we choose to use the Toy-car machine type only. For training, normal sound samples of seven different toy-cars are provided, each having 1000 samples mixing the machine sound with environmental noise.

**Model** While several models are openly available from the competition, the competition itself did not focus on TinyML, and only a few of these fit our target model size. Among these, we have selected the AE based model also used in the reference implementation of DCASE2020 for multiple reasons. First, it is the reference for a large part of the literature around the audio anomaly detection problem

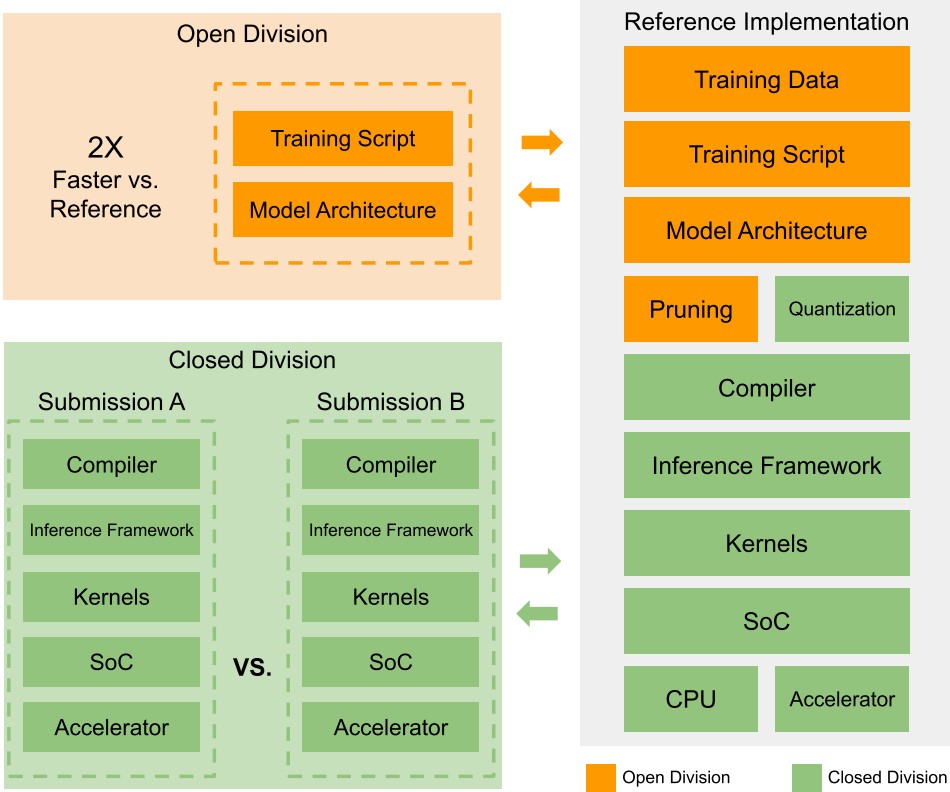

Figure 2: The modular design of MLPerf Tiny enables both the direct comparison of solutions and the demonstration of an improvement over the reference. The reference implementations are fully implemented solutions that allow individual components to be swapped out. The components in green can be modified in either division and the orange components can only be modified in the open division. The reference implementations also act as the baseline the results.

and thus a well known baseline. Second, it enhances the benchmark suite with a new model type based entirely on FC layers. The model has input and output sizes of 640. Both the encoder and decoder are made of four 128 unit FC layers with BatchNorm applied during the training and with ReLU activation, while the bottleneck layer is of size 8. The model itself is not applied directly on the 10 second audio. The audio is pre-processed into a log-mel-spectrogram with 128 bands and a frame size of 32 ms. Then, the model is used repeatedly over a sliding window of five frames (hence the 640 input size), and the MSE of the resulting reconstruction error is averaged over the central 6.4 second part of the spectrogram providing an overall anomaly score.

**Quality Target** The output of the autoencoder is an anomaly score, which assumes a further threshold to be set before the binary normal/anomalous decision can be made. For this reason the parameterless AUC-ROC (Area Under The Curve, Receiver Operating Characteristics) fits better for the quality evaluation for this problem than metrics requiring the selection of a threshold. To ensure models have generalization characteristics, we have selected an evaluation dataset composed of normal and anomalous sounds from four different machines totalling 248 samples. On this set, the fp32 version of the reference model achieves an AUC of 0.88 while the AUC after quantization is 0.86. Based on these two numbers, we've set the threshold for the benchmark to AUC 0.85.

## 5 Run Rules

In this section we describe the modular design of the benchmark harness, as well as the closed and open benchmark divisions, and the overall run rules of the benchmark suite to ensure reproducibility.

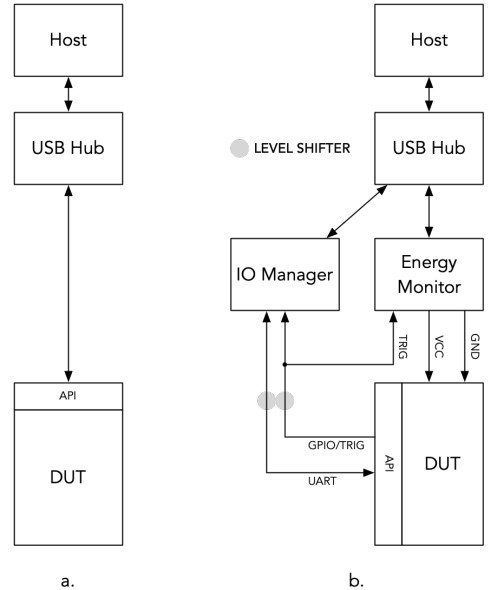

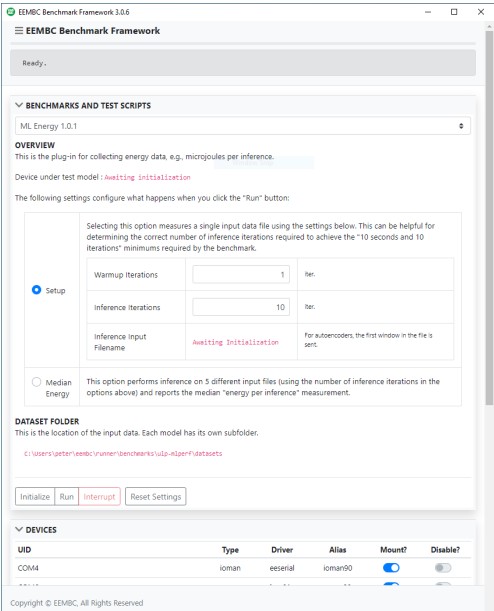

Figure 3: The two configuration modes of the benchmark framework for (a.) latency and accuracy measurement, or (b.) energy measurement.

Figure 4: The graphical user interface (GUI) for the benchmark runner.

## 5.1 Modular Design

TinyML applications often require cross stack optimization to meet the tight constraints. Hardware and software users can provide value by targeting specific components of the pipeline, like quantization, or offer end-to-end solutions. Therefore, to enable users to demonstrate the competitive advantage of their specific contribution, we employ a modular approach to benchmark design. Each benchmark in the suite has a reference implementation that contains everything from training scripts to a reference hardware platform. This reference implementation not only provides a baseline result, it can be modified by a submitter wishing to show the performance of a single hardware or software component. Figure 2 illustrates the modular components of a MLPerf Tiny reference implementation.

## 5.2 Closed and Open Divisions

MLPerf Tiny has two divisions for submitting results: a stricter closed division and a more flexible open division. A submitting organization can submit to either or both divisions. This two division design allows the benchmark to balance comparability and flexibility. Figure 2 illustrates which components of the reference benchmark can be modified in which division. If only the components shaded in green are modified then the submitter can submit to the open division. If any of the components shaded in orange are modified then the submitter must submit to the open division.

The closed division enables a more direct comparison of systems. Submitters must use the same models, datasets, and quality targets as the reference implementation. The closed division permits post training quantization using the provided calibration datasets, but prohibits any retraining or weight replacement. Figure 2 demonstrates how submissions to the closed division allow the comparison of an inference stack to another in a controlled setting.

The open division is designed to broaden the scope of the benchmark and allow submitters to demonstrate improvements to performance, energy, or accuracy at any stage in the machine learning pipeline. The open division allow submitters to change the model, training scripts, and dataset. A submission to the open division will still use the same test dataset to benchmark the accuracy but is not required to meet the accuracy threshold, which allow submitters to demonstrate tradeoffs between accuracy, latency, and energy consumption. Each submission to the open division must document how it deviates from the reference implementation. Figure 2 demonstrates how submissions to the open division allow users to demonstrate the specific value add of their product.

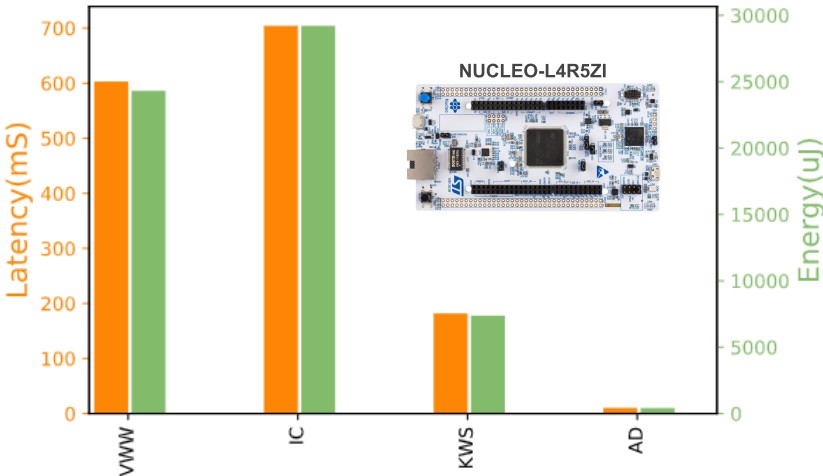

Figure 5: The energy and latency results of the reference implementations. Each reference implementation was run on the NUCLEO-L4R5ZI board which is shown in the top right.

### 5.3 Measurement Procedure

The platforms targeted by TinyML typically do not contain the resources required to run a complete benchmark locally, e.g., file IO, standard input and output, interactivity, etc. As a result, a benchmark framework (Figure 3) is required to facilitate controlling and measuring the device under test (DUT). The benchmark framework used to implement the MLPerf Tiny benchmark is based on EEMBC's software development platform. The resulting program coordinates execution of the benchmark's behavioral model among the various components in the system as described in the Appendix. The framework's measurement procedure is as follows:

1. Latency – Perform this five times: download an input stimulus, load the input tensor (converting the data as needed), run inference for a minimum of 10 seconds and 10 iterations, measure the inferences per second (IPS); report the median IPS of the five runs as the score.

2. Accuracy – Perform a single inference on the entire set of validation inputs, and collect the output tensor probabilities. The number of inputs vary, depending on the model. From the individual results, compute the Top-1 percent and the AUC. Each model has a minimum accuracy that must be met for the score to be valid.

3. Energy – Identical to latency, but in addition to measuring the number of inferences per second, measure the total energy used in the timing window and compute micro-Joules per inference. The same method of taking the median of five measurements is used.

Results are stored in a folder and can be reloaded again. An energy viewer allows the user to examine the energy trace. Figure 5 illustrates the reference implementation results. The four benchmarks cover a wide scope in terms of latency and energy and each reference meets the minimum accuracy.

## 6 Benchmark Assessment

In this section we assess the success of the benchmark at meeting the needs of the community. The suite is designed to provide standardization and compatibility while enabling a diverse set of organizations to submit results.

### 6.1 Submissions

The MLPerf Tiny benchmark suite accepts results from submitting organizations twice a year. The submitting organizations must implement the benchmarks on their hardware/software stack and submit their results and implementations at the deadline. All results are then transparently peer-reviewed by the group of submitters and a review committee to ensure they conform to the rules.

| Division | Dataset | Training | Model | Numerics | Framework | Hardware | Demonstrates |
|---|---|---|---|---|---|---|---|
| Closed | X | X | X | INT-8 PTQ | TensorFlow Lite Micro | ARM MCU | Baseline performance results on the reference platform. |
| Closed | X | X | X | INT-8 PTQ | TensorFlow Lite Micro | RISC-V MCU | Performance of a RISC-V microcontroller customized for neural network inference. |
| Closed | X | X | X | FP-32 & INT-8 PTQ | LEIP Framework | RasPi 4 | Capabilities of a software-only optimization toolchain that is agnostic of the hardware. |
| Closed | X | X | X | INT-8 PTQ | Syntiant TDK | Neural Network Accelerator | Ultra-low power hardware efficiency for running deep neural networks. |
| Open | X | QKeras | ✓ | Int-6/8 QAT | HLS4ML | FPGA | Rapid end-to-end development of machine learning accelerators on reconfigurable fabrics. |

Table 2: Summary of the MLPerf Tiny v0.5 round of submissions. The submissions are diverse and illustrate the ability of the benchmark to demonstrate an advantage in a variety of ways. The X indicates no modification was made from the reference. The check mark indicates a modification. PTQ refers to post training quantization and QAT refers to quantization aware training.

The first round of submissions to MLPerf Tiny (which took place in June 2021) are summarized in Table 2. The results from the first round submissions are currently available at `https://mlcommons.org/en/inference-tiny-05/`. Each submission is publicly available on GitHub with instructions on how to reproduce each result at `https://github.com/mlcommons/tiny_results_v0.5`.

## 6.2 Insights

The v0.5 submissions were diverse and demonstrated the desired flexibility of the benchmark suite. There were submissions to the open and closed divisions, as well as from hardware and software vendors. Each submitter had a specific element of their submission they wished to demonstrate and the benchmark suite was able to accommodate this diverse set of goals due to its modular design.

The submissions indicate general trends in TinyML. For example, the most common numerical format is 8-bit integer for inference as it offers a performance boost with little impact to the model accuracy. ML frameworks range from open source interpreters (TFLite Micro) to hardware specific inference compilers, indicating that there is still often a trade-off between optimization and portability. There were results collected on a wide variety of hardware platforms, including MCUs, accelerators, and FPGAs. The re-configurable hardware (FPGA) is able to utilize variable precision models for increased performance. The power consumption of these platforms ranged from µWatts to Watts.

More specifically, one organization used the benchmark to demonstrate that their SDK developer tools are hardware agnostic, easy to use and uniquely designed to optimize neural network inferences for compute, energy and memory, while preserving algorithmic accuracy. Another organization, a hardware vendor, used the benchmark to demonstrate outstanding performance on their NN accelerator, enabled by their novel microarchitecture design and optimized datapath, which avoids wait-states and maintains high computational efficiency. An academic research institute used the benchmark to demonstrate AI compute capability and potential applications for RISC-V-based AI micro-controllers. RISC-V is a free, open standard instruction set architecture (ISA) [3]. Finally, a multi-disciplinary team of scientists and engineers showcased their "hls4ml" (high-level synthesis for ML) open-source workflow. It was designed to enable researchers and engineers to codesign optimized neural networks for efficient dataflow architectures on a multitude of accelerator hardware platforms. Originally developed for the Large Hadron Collider to make ultrafast sub-microsecond ML inference, the workflow aims to serve the wider ML community to accelerate both the design process and neural network implementation across a broad range from low-power to high-performance devices.

These results indicate a snapshot of this emerging field but future submission rounds can be used to indicate the evolution of TinyML over time. For instance, despite a general trend in AI towards data-centric design, none of the submissions in the first round modified the training dataset. While we anticipate this will shift in future versions, TinyML design is still largely focused on models, frameworks, and hardware. To this end, as demonstrated, the benchmark meets a variety of needs.

## 7 Impact

The benchmarks have already acted as a standard set of tasks for TinyML research [4] and have been made into public projects on a TinyML development platform [16]. The benchmark will standardize the nascent field of TinyML and enable future progress through competition and comparability. TinyML as a field has the potential to democratize AI by removing the barrier of expensive hardware and can preserve privacy by keeping user's data on the device that captures it. At the same time, the technology could be misused to more efficiently track and monitor unwilling individuals. Furthermore, because the devices themselves are inexpensive, TinyML can lead to increased electronic waste. By establishing a collaborative community, MLPerf Tiny can aid in the creation of standards for the responsible deployment of TinyML and mitigate the potential negative impacts of the technology.

## 8 Limitations

**New benchmarks & Long-term stability** MLPerf Tiny will continue to evolve to reflect the needs of the community. This will include new benchmarks that target new applications domains, such as wearables, medical devices, and environmental monitoring. While new benchmarks are being added and existing ones are evolved, it is also important to keep in mind that such a benchmark serves both as the comparison of state-of-the-art and for tracking historical progress. This latter goal requires benchmarks to be stable over time, eventually providing a multi-year perspective into the evolution of the field. Keeping the benchmark open source helps long-term reproducibility, while we also envision a subset of benchmarks to be kept long-term stable for this purpose. Also, thanks to MLCommons (mlcommons.org), a non-profit open engineering organization that hosts and develops the MLPerf benchmarks, the MLPerf Tiny benchmark will continue to be supported through future generations.

**Streaming inputs & Pre-processing** Time-domain tasks, such as the keyword spotting and anomaly detection benchmarks, typically involve continuously streaming inputs. Information from previous time steps can be exploited to improve the performance-efficiency tradeoff. However, limited bandwidth between the test runner (running on e.g. a PC) and the DUT makes it difficult to recreate a streaming scenario without adding delays incurred by data transfer. The choice of whether to include pre-processing in the measured benchmark can also have distorting effects on the results. Excluding feature extraction from measurement while allowing submitter-selected variations in feature extraction creates the possibility of a degenerate case where an entire model up to the penultimate layer is defined as "feature extraction". Rigidly defined feature extraction precludes joint optimization over feature and model architecture that can be critical in the constrained systems targeted by this benchmark suite. Inclusion of feature extraction in the benchmark brings its own subtle complexities. Operating on a single discrete audio input in a non-streaming benchmark, one second of audio would involve one inference cycle and 40 feature extraction cycles, over-emphasizing the cost of feature extraction. In future versions, we aim to widen the scope of the benchmarks to include pre-processing as well.

**Extended coverage of layer types and model architectures** The closed division of the current benchmark suite includes models based mainly on FC and CNN layers in a variety of well known architectures, allowing only open division submissions to deviate from these. While our current suite provides a reasonable baseline that can be implemented on most HW platforms and openness to more experimental submissions, future benchmarks may include reference implementations using additional architectures e.g. RNNs.

## 9 Conclusion

The field of TinyML is poised to drive enormous growth within the IoT hardware and software industry. However, measuring the performance of these rapidly proliferating systems and comparing them in a meaningful way presents a considerable challenge; the complexity and dynamicity of the field obscure the measurement of progress and make embedded ML application and system design and deployment intractable. To enable more systematic development while fostering innovation, we need a fair, replicable, and robust method of evaluating TinyML systems. Developed as a collaboration between academia and industry, MLPerf Tiny is a suite of benchmarks for assessing the energy, latency, and accuracy of TinyML hardware, models, and runtimes. The benchmark suite and reference implementations are open-source and available at `https://github.com/mlcommons/tiny`.

## Acknowledgments

MLPerf Tiny is the work of many individuals from multiple organizations. In this section, we acknowledge all those who helped produce the first set of results or supported the overall benchmark development. This work was also sponsored in part by the ADA (Applications Driving Architectures) Center, a JUMP Center co-sponsored by SRC and DARPA.

**Amazon** Amin Fazel

**Arizona State University** Jae-sun Seo

**California State Polytechnic University** Robert Hurtado

**CERN** Nicolo Ghielmetti

**Cisco Systems** Xinyuan Huang

**Columbia** Shvetank Prakash (Harvard)

**dividiti** Anton Lokhmotov

**Fermilab** Benjamin Hawks and Jules Muhizi (Harvard)

**Google** Ian Nappier, Dylan Zika, and David Patterson (UCB)

**Harvard** Max Lam and William Fu

**KU Leuven** Marian Verhelst (IMEC)

**ON Semiconductor** Jeffrey Dods

**Peng Cheng Lab** Yang Shazhou and Tang Hongwei

**Reality AI** Jeff Sieracki

**Renesas** Osama Neiroukh

**Silicon Labs** Dan Riedler

**STMicroelectronics** Mahdi Chtourou

**Synopsys** Dmitry Utyansky

**Syntiant** Mohammadreza Heydary and Rouzbeh Shirvani

**UCSD** Rushil Roy

**University of York** Poonam Yadav

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

# A    Benchmark Framework

The benchmark framework coordinates execution of the benchmark's behavioral model among the various components in the system as described in the following sections.

## A.1    Framework Hardware

In its most basic form, the framework consists of a host PC, a binary runner application GUI, the DUT, and a thin shim of firmware on the DUT that adheres to the framework communication protocol.

The framework for this particular benchmark provides two hardware configurations: one for measuring latency and accuracy, and the other for measuring energy (see Figure 3). The former mode is a simple connection between the host PC and the DUT through a serial port. The latter configuration expands the framework to include an electrical-isolation proxy (IO Manager), and an energy monitor (EMON), which supplies and measures energy consumption. Both configurations share the same process for initializing the DUT, loading the input test data, triggering the inference, and collecting results. The only difference is the energy configuration must initialize and manage the IO Manager and EMON hardware as well.

Two framework configurations are provided because energy configuration is a more complex setup, and energy scores may not be desired.

In the performance configuration, the Host PC talks directly to the DUT. In energy configuration, the IO Manager passes commands from the host software to the DUT through a serial port proxy. This proxy maintains a resilient serial connection state regardless of power cycling, and level shifters provide voltage domain conversion because the IO Manager GPIOs run at 5 Volts, whereas the DUT GPIO may vary. This configuration electrically isolates the DUT. For this framework version, the IO Manager is deployed as an Arduino UNO with its own custom firmware.

The energy monitor supplies and measures energy. The runner contains three EMON drivers for the following hardware: the STMicroelectronics LPM01A, the Jetperch Joulescope JS110, and the Keysight N6705. Regardless of the EMON used, it must supply one channel for the DUT and the other for the level shifters, the power used by level-shifters is not included in the total energy score because it is a cost associated with the framework and not the DUT. Only the power delivered to the core is measured. Furthermore, only one power supply is allowed to power the core; no additional energy supplies, such as batteries, supercapacitors, or energy harvesters may be used to defeat the measurement. Some energy monitors provide settings to increase the sample frequency or change the voltage. The runner GUI exposes these options to the user, so that the user can measure energy at a lower voltage. Note that the tradeoff between performance and energy can be dictated by the voltage, as higher voltage is often required to run at higher frequencies, and on-board SMPS conversion efficiency may vary considerably. This is one of the key insights of the benchmark: performance vs. energy tradeoffs.

The configurable sampling frequency is the rate at which data is returned from the EMON, which is much lower than the actual ADC sampling rate for all three devices. Since these internal sampling rates are typically well-within the time constant of the decoupling DUT's power delivery decoupling capacitance, the rate has no impact on the score.

## A.2    Framework Firmware

To provide consistency between the benchmark framework and the DUT implementation, a simple API is defined in C code that runs on the DUT. The DUT must provide two basic functions to interface to the host runner: a UART interface for receiving commands and returning status, and a timestamp function. In performance mode, the timestamp function is a local timer with a resolution of at least one millisecond (1 kHz); in energy mode, the timestamp function generates a falling edge GPIO signal which is used to trigger an external timer and synchronize data collection with the energy monitor. The API C code also provides functions to load data into a local buffer, translate and copy that data to the input tensor, trigger an inference, and read back the predictive results.

All of this functionality is partitioned into boilerplate code that does not change, or internally implemented, and code that has to be ported to the particular platform, called submitter implemented.

The submitter implemented code is the API that connects to the particular SDK, which may consist of optimized MCU libraries, or interface to hardware accelerators.

The API implemented by the submitter requires:

1. A timestamp function which prints a timestamp with a minimum resolution of one millisecond, or generates an open-drain, GPIO toggle, depending on whether the DUT is configured latency or energy measurement.

2. UART Tx and Rx functionality, for communicating with the framework software.

3. A function for loading the input tensor with data sent down by the framework runner UI.

4. A function for performing a single inference.

5. A function for printing the prediction results.

The first two API functions provide external triggering and generic communication support; the latter three implement the minimum requirements to perform inference. Once these functions have been implemented, the framework software can successfully detect and communicate with the DUT.

The internally implemented firmware connects the API function calls in such a way that the framework software can send down an input dataset and instruct the firmware to perform a given number of iterations. The results are then extracted from the DUT (or the energy monitor) by the framework software.

## A.3   Framework Software

The benchmark framework includes a Host PC GUI application called the runner. The runner allows the user to perform some basic setup and configuration, and then executes a pre-defined command script which activates the different hardware components required to execute the benchmark.

The runner is needed for three main reasons:

1. To provide a consistent benchmark interface to the user.

2. To standardize the execution of the benchmark, including measurements

3. To facilitate downloading the large number of input files required for accuracy measurements, since the typical target platform has less than a megabyte of flash memory.

The runner also provides visualization of the energy data, and feedback that can be useful for debugging framework connectivity.

The runner GUI, for the energy configuration, is shown in Figure 4.

The runner detects and initiates a handshake with detected hardware. After initialization, the test is started by clicking a button on the GUI, and a series of asynchronous commands are issued. These commands use the DUT API firmware to load input data and perform measurements. After the test completes, the runner collects the scores from the DUT and presents them to the user. The input stimuli files are located in a directory on the host PC, and are fed into the DUT depending on the measurement.

The measurement procedure is as follows:

1. Latency – Perform the following five times: download an input stimulus, load the input tensor (converting the data as needed), run inference for a minimum of 10 seconds and 10 iterations, measure the inferences per second (IPS); report the median IPS of the five runs as the score.

2. Accuracy – Perform a single inference on the entire set of validation inputs, and collect the output tensor probabilities. The number of inputs vary, depending on the model. From the individual results, compute the Top-1 percent and the AUC. Each model has a minimum accuracy that must be met for the score to be valid.

3. Energy – Identical to latency, but in addition to measuring the number of inferences per second, measure the total energy used in the timing window and compute micro-Joules per inference. The same method of taking the median of five measurements is used.

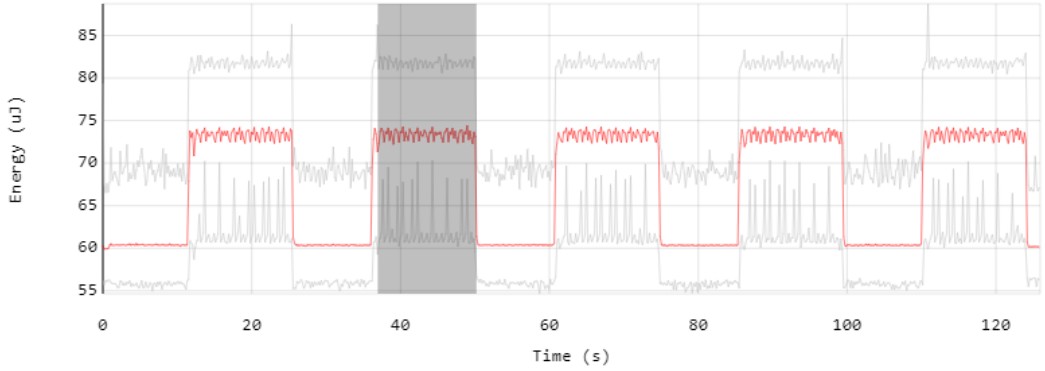

Figure 6: Energy Viewer

Results are stored in a folder and can be reloaded again. An energy viewer allows the user to examine the energy trace (Figure 6)

