# OpenReview forum: "MLPerf Tiny Benchmark"
_NeurIPS.cc/2021/Track/Datasets_and_Benchmarks/Round1 — NeurIPS 2021 Datasets and Benchmarks Track (Round 1)_

### Official Review · Reviewer_YyGE · 2021-07-04
**Review of "MLPerf Tiny Benchmark"**

**Rating:** 6
**Confidence:** 2
**Clarity:** This paper is mostly well-written.

**Strengths:**

+ The contributions of the new benchmark on tiny ML are significant since the widespread use of ML models in smart devices require the fair comparisons between different methods and models. Establishing a new benchmark in this field may be valuable.
+ The benchmark is relevant to a broad research community, and especially from the industry perspective.
+ The benchmark is open-sourced and will be maintained in the future.

**Weaknesses:**

I'm not an expert in this area. From my perspective, I have few concerns about the benchmark.

My biggest question is about the evaluation. Although the authors have introduced many aspects of the benchmark, I do not see many evaluation results done in this benchmark. As there may be many existing methods/models performing tiny ML, it is expected to conduct comprehensive studies to compare the existing methods. Based on the evaluation, could you have some findings or the limitations of existing methods?

For Figure 1, can you provide more illustrations about the connections to the benchmark? E.g., how do you deal with each level of Figure 1 in your benchmark.

**Additional Feedback:**

Please address the concerns about the evaluation.

**Correctness:**

This benchmark includes various tasks, all of which use the public datasets. There is no data collection process in this benchmark. However, this benchmark only introduces the techniques without many experiments. So the experiment design cannot be judged correctly.

**Documentation:**

The source code of the benchmark is provided. However, there are no experiments nor benchmark results.

**Ethics:**

I did not see any ethical issues.

**Relation To Prior Work:**

This paper claimed that it is the first benchmark. The related work is introduced thoroughly (to me).

**Summary And Contributions:**

This paper focuses on the field on tiny machine learning which are used in ultra-low-power devices. The first industry-standard
 benchmark is introduced in this paper named "MLPerf Tiny". The details of the new benchmark are presented in various aspects, including tasks, datasets, models, evaluation process, and system design.

This paper makes the following contributions:

1. A new benchmark on tiny machine learning is established.
2. The related work is introduced thoroughly. The motivation of building such a benchmark is clear.
3. The benchmark design is detailed with a modular implementation.
4. It includes various tasks including image classification, keyword spotting, etc.

---

### Official Review · Reviewer_F9Bz · 2021-07-04
**Good paper**

**Rating:** 6
**Confidence:** 2
**Correctness:** Yes.
**Clarity:** Yes, the paper is well written.

**Strengths:**

This paper presents the first industry-standard benchmark suite for ultra-low-power tiny machine learning systems, it could push the development for compact model design.

**Weaknesses:**

Specific experiments are lacking, such as the accuracy, latency, and energy of the ML models (Mobilenet，MicroNets，RNN, AutoEncoder) on MLPerf Tiny Benchmark.

**Additional Feedback:**

It would be better if there are more specific experiments, such as the accuracy, latency, and energy of the ML models (Mobilenet，MicroNets，RNN, AutoEncoder) on MLPerf Tiny Benchmark.

**Documentation:**

Yes. But more checks maybe needed.

**Relation To Prior Work:**

Yes, this paper compares the proposed dateset with previous contributions.

**Summary And Contributions:**

This paper presents MLPerf Tiny, the first industry-standard benchmark suite for ultra-low-power tiny machine learning systems. MLPerf Tiny measures the accuracy, latency, and energy of machine learning inference to properly evaluate the tradeoffs between systems.

---

### Official Review · Reviewer_k8ef · 2021-07-07
**Standardized and Diverse Benchmarks for TinyML**

**Rating:** 7
**Confidence:** 4
**Clarity:** The paper is well written.

**Strengths:**

1. The benchmark contains a wide range of tasks, despite lacking NLP-related ones. The tasks are also oriented at low-resource environments and applications can be found in real life (e.g., single person detection for doorbells).
2. The measurement considers latency, energy and accuracy. The use of a quality threshold instead of the commonly used tradeoff curve comparisons greatly simplifies the comparison.
3. The benchmark has two divisions: an open division that allows the change of model, data, training, and a closed division that focuses more on post-training optimization. This allows different methods that focus on different parts of the whole optimization process to show their benefits more clearly.
4. The challenges for constructing a standardized TinyML benchmark are well summarized in the related work section.

**Weaknesses:**

The running and submission process to the benchmark, as documented in the Appendix, requires significantly more effort than a typical benchmark submission, where submitters usually only submit the test results. This can limit the number of submitters. However, this is perhaps a tradeoff that has to be made for more comparability for the widely heterogenous TinyML works.

It would also be nice to include a summary of the execution process in the main paper instead of only documenting in Appendix.



**Additional Feedback:**

Please see strengths and weaknesses.

**Correctness:**

Yes. The benchmark is constructed from previously published datasets, with adaptations made for low-resource and simplicity considerations. The evaluation and experiment design are well documented in paper.

**Documentation:**

Yes.

**Ethics:**

The potential ethical impact was discussed in paper.

**Relation To Prior Work:**

Yes.

**Summary And Contributions:**

The paper collects and adapts previously published datasets to construct a standardized benchmark for TinyML methods, which previously suffer from poor comparability due to software/hardware heterogeneity. The tasks include keyword spotting, visual wake words, image classification, and anomaly detection. The benchmark has two divisions, one allowing any changes to the pipeline and one only allowing post training optimization. It measures accuracy, latency and energy use.

---

### Author Response · Authors · 2021-07-12
**MLPerf Tiny Benchmark - Revision and Response**

We would like to thank the reviewers for their comments and suggestions. We have updated the paper to include the suggested improvements. We have also included the v0.5 results which have recently been made publicly available.

**The changes to the paper are as follows:**
- Section 5.3 has been expanded to include more information about the execution process
- Table 2 has been added to summarize the v0.5 round of submissions
- Section 6 has been reworked to include more information and evaluation
-- 6.1 includes links to the v0.5 results and instructions to reproduce each result
-- 6.2 has been added to provide insight on the v0.5 round of submissions, including how the benchmark suite demonstrated the desired flexibility and specificity.
- Section 8.1 has been added which discusses the long term support of the benchmark suite

This amounts to a full additional page of information and evaluation.

**We will now directly address the reviewers comments:**

> The running and submission process to the benchmark requires significantly more effort than a typical benchmark submission (Reviewer k8ef)

In order to provide a fair benchmark that measures real on-device performance, the process has to be somewhat rigorous. We reduced the effort required by 1) developing detailed reference implementations that act as starting points for submitters and 2) making energy measurement optional to reduce implementation effort. Additionally, this is something we aim to improve in future versions of the benchmark suite.


> Include a summary of the execution process in the main paper (Reviewer k8ef)

Section 5.3 has been expanded to include more information about the execution process


> Specific experiments/evaluation results are lacking (Reviewers F9Bz & YyGE)

 We have included a summary and link to the version 0.5 round of submission to the MLPerf Tiny Benchmark. We have also included a link to the instructions on how to reproduce each result. Section 6 has been reworked and extended to include more evaluation of the benchmark and discussion of the first set of results. We received a diverse set of submissions, demonstrating the performance of a variety of models, frameworks, and hardware platforms.


> For Figure 1, provide more illustrations about the connections to the benchmark (Reviewer YyGE)

The newly added Table 2 illustrates the components each submitter selected at each layer in the stack. Additionally, section 5.1 discusses how the benchmark was designed to accommodate the heterogeneity of the ML system stack. We select four benchmarks that attempt to cover the variety of sensors, applications, datasets, and models. The benchmark itself allows submitters to demonstrate their own implementations of the remaining layers, meaning we can show the value in each component.


**Relevant Links:**

The v0.5 results are available here: https://mlcommons.org/en/inference-tiny-05/

The code/instructions to reproduce are available here: https://github.com/mlcommons/tiny_results_v0.5

The benchmark has already generated interest from the public and the press: https://www.google.com/search?q=mlperf+tiny

---

### Decision · Program_Chairs · 2021-07-26

**Decision:**

Accept

**Comment:**

The paper proposes a benchmark for machine learning models running on low-power devices. All reviewers saw value in the proposed benchmark, but initially had some minor concerns and requested more details about how to run the benchmark, wanted to see more evaluation results, and wanted to see more discussion about connections between different layers of the TinyML stack (Figure 1). The authors were able to address these concerns through their responses and revisions to the paper, and in the end all reviewers voted to accept the paper. Congratulations on having your paper accepted to the NeurIPS 2021 Datasets and Benchmarks Track! The authors are encouraged to use the additional space in the camera-ready version of the paper to continue refining the language and presentation in light of the new additions suggested by reviewers.